TCellR2Vec: efficient feature selection for TCR sequences for cancer classification

Tayebi Zahra
Ali Sarwan
Patterson Murray mpatterson30@gsu.edu
Computer Science, Georgia State University , Atlanta , GA , United States of America
Raza Khalid
Electronic publication date: 2024 Nov 4
Publication date: 2024
Volume: 10
Electronic Location ID: e2239
Received 2024 Mar 6; Accepted 2024 Jul 14
Copyright: ©2024 Tayebi et al.
Copyright year: 2024
Copyright holder: Tayebi et al.
License: This is an open access article distributed under the terms of the Creative Commons Attribution License, which permits unrestricted use, distribution, reproduction and adaptation in any medium and for any purpose provided that it is properly attributed. For attribution, the original author(s), title, publication source (PeerJ Computer Science) and either DOI or URL of the article must be cited.
License URL: https://creativecommons.org/licenses/by/4.0/

Keywords: Cancer, TCR sequence, Feature selection, Classification

Funding: Brain and Behavior (BB) fellowship Molecular Basis of Disease (MDB) fellowship Georgia State University startup fund Zahra Tayebi was funded by Brain and Behavior (BB) fellowship. Sarwan Ali was funded by Molecular Basis of Disease (MDB) fellowship. Murray Patterson is backed by a Georgia State University startup fund. The funders had no role in study design, data collection and analysis, decision to publish, or preparation of the manuscript.

==============================
Cancer remains one of the leading causes of death globally. New immunotherapies that harness the patient’s immune system to fight cancer show promise, but their development requires analyzing the diversity of immune cells called T-cells. T-cells have receptors that recognize and bind to cancer cells. Sequencing these T-cell receptors allows to provide insights into their immune response, but extracting useful information is challenging. In this study, we propose a new computational method, TCellR2Vec, to select key features from T-cell receptor sequences for classifying different cancer types. We extracted features like amino acid composition, charge, and diversity measures and combined them with other sequence embedding techniques. For our experiments, we used a dataset of over 50,000 T-cell receptor sequences from five cancer types, which showed that TCellR2Vec improved classification accuracy and efficiency over baseline methods. These results demonstrate TCellR2Vec’s ability to capture informative aspects of complex T-cell receptor sequences. By improving computational analysis of the immune response, TCellR2Vec could aid the development of personalized immunotherapies tailored to each patient’s T-cells. This has important implications for creating more effective cancer treatments based on the individual’s immune system.

Introduction

According to the World Health Organization (WHO), cancer is one of the leading causes of death worldwide (World Health Organization, 2023). In 2022, in the United States alone, an estimated 236,740 new cases of lung cancer are expected, resulting in approximately 130,180 deaths (Siegel et al., 2022). Disease variations across different regions, the influence of available medical resources, and various socio-economic factors have collectively affected the effective management of this disease (Chhikara & Parang, 2023). However, effective treatment can cure many types of cancer (World Health Organization, 2023). For instance, glioblastoma is an aggressive form of brain cancer that arises from glial cells in the brain or spinal cord (Wirsching & Weller, 2017). It is characterized by a high degree of heterogeneity and resistance to current treatment options, leading to a poor prognosis for patients (Soeda et al., 2015). Lung cancer is another leading cause of cancer-related deaths worldwide and it can be classified into two main types: small cell lung cancer and non-small cell lung cancer (Minna, Roth & Gazdar, 2002; Lahiri et al., 2023). Another extremely deadly cancer type is Melanoma, which is a type of skin cancer that is caused by the uncontrolled growth of melanocytes, the cells that produce pigment in the skin (Houghton & Polsky, 2002). Immunotherapy has shown promise in reducing the risk of melanoma recurrence post-surgical resection, and in enhancing survival rates among patients with unresectable versions of the disease. disease (Knight, Karapetyan & Kirkwood, 2023). Also, pancreatic cancer arises from the cells in the pancreas and is known for its aggressive behavior and low survival rate (Kleeff et al., 2016). Based on data from the Surveillance, Epidemiology, and End Result Program (SEER), approximately 57,600 new cases of lung cancer were diagnosed in 2020, resulting in 47,050 deaths (Howlader et al., 2020). Whether a tumor can be removed with surgery (resectable) affects treatment options and how well patients do (Millikan et al., 1999; Kolbeinsson et al., 2023). Another example is a rare type of bone cancer called Osteosarcoma that develops in the cells that form the bone and can cause severe pain and disability (Ta et al., 2009). Osteosarcoma exhibits complex heterogeneity and abnormal production of immature osteoid matrix (Liu et al., 2022). Current treatments for Osteosarcoma struggle to eliminate all cancer cells, particularly those that have spread (metastatic) or are circulating in the bloodstream, (Li et al., 2021; Lamhamedi-Cherradi et al., 2021). These cancers represent a significant challenge to healthcare professionals and researchers alike, highlighting the urgent need for effective treatment options.

Traditional cancer treatments such as chemotherapy and radiation therapy are often associated with significant side effects (MacDonald, 2009; Schirrmacher, 2019), and there is a need for more personalized and targeted treatments. Recently, immunotherapy has emerged as a promising approach to treat cancer by utilizing the patient’s own immune system to target cancer cells (Lizée et al., 2013; Kciuk et al., 2023). One key component of the immune system is T-cells which are a type of white blood cell (Beshnova et al., 2020; Speiser et al., 2023) and T-cell receptors (TCRs) which are proteins found on the surface of T-cells. The diversity of TCRs allows them to recognize and bind to specific proteins or peptides presented by cancer cells, as depicted in Fig. 1 (Raskov et al., 2021; Nikolich-Žugich, Slifka & Messaoudi, 2004). Almost all cancer immunotherapies approved by the US Food and Drug Administration (FDA) work by activating and expanding T cells that express TCRs capable of recognizing tumor antigens (Rosenberg, 2014; Ribas & Wolchok, 2018).

Figure 1 T-cells bind to specific peptides presented by cancer cells.

TCR sequencing has emerged as a crucial tool for comprehending the immune response to cancer and developing personalized cancer treatments (Pai & Satpathy, 2021). The TCR sequencing data provides valuable information about each TCR sequence like the type and location of the antigen that the T-cell has identified (Klebanoff et al., 2023; Saotome et al., 2023). Understanding the complexity and diversity of these sequences has created a significant need for feature selection methods. Feature selection methods are techniques used to select a subset of relevant and informative features from a larger set of input variables or features (Li et al., 2017). Several embedding generation methods generate a lower-dimensional representation of the data and can be used as a form of feature selection.

Machine learning-based embedding generation methods can capture information about protein sequences that can be used for classification tasks (Stein, Jaques & Valiati, 2019; Bukhari et al., 2022; Bukhari et al., 2021). However, due to the specificity of the cancer-related protein sequences we are working with, general embedding generation methods such as one-hot encoding (OHE) and spaced K-mers may not be effective in fully capturing the complexity and diversity of these sequences. Deep learning methods, such as recurrent neural networks (RNNs) and convolutional neural networks (CNNs), are capable of handling the complexity and diversity of protein sequences by learning complex relationships between sequence features (Lyu et al., 2023). RNNs can process sequences by using recurrent connections to propagate information from one step to the next (Tsukiyama et al., 2021). Also, CNNs can be applied to sequences by treating them as a one-dimensional signal and using convolutional filters to extract features (Vosoughi, Vijayaraghavan & Roy, 2016). However, using deep learning methods to generate embeddings is computationally intensive and requires a large amount of computational resources to train and run (Heinzinger et al., 2019). This can be a challenge for researchers who may not have access to high-performance computing resources.

In this study, we used machine learning approaches along with an informative feature selection method to train these models effectively and faster while also improving their accuracy. This study showcases TCellR2Vec’s effectiveness in selecting features for cancer classification from TCR sequences. Beyond this, it has broader implications. TCellR2Vec aids in identifying tumor-reactive T-cell receptors and neoantigens, leading to personalized cancer immunotherapies like tailored vaccines and cell transfer therapies. It also helps monitor immune responses during treatment, discover biomarkers for prognosis, and understand tumor-immune interactions. Integrating TCellR2Vec with neoantigen prediction improves vaccine development and enhances cancer immune evasion understanding. Our contributions can be summarized as follows:

1. We introduce a novel feature selection method named TCellR2Vec, which generates embeddings that can be used as input to machine learning classifiers to improve supervised analysis.

2. To provide a numerical representation of the TCR sequences, we extract multiple features, including amino acid sequence compositions, the length of the CDR3, hydrophobicity, charge distribution, average motif similarity score, Shannon entropy, and Simpson index, using a real dataset of around 50K TCR sequences. These features are utilized to generate input for the classification models.

3. To ensure that our classification methods were not prone to overfitting or slow performance, we carefully selected six different embedding generation methods from the literature and aggregated (concatenated) them with our proposed embedding method (TCellR2Vec) to provide a diverse set of embeddings with different properties and characteristics.

4. We assess the effectiveness of our approach by employing various metrics and comparing it to other state-of-the-art methods without using our extracted feature vectors. Our findings indicate that merging baselines with our feature extraction method improves their evaluation metrics like predictive accuracy and efficiency for classification.

Based on the aforementioned considerations, the paper is organized as follows. The ‘Related Work’ section offers an overview of prior research on the problem addressed in this study. Following that, the proposed approach section introduces our method. The ‘Experimental Setup’ section details the dataset and experimental configuration utilized to assess our proposed approach and baselines. The results of the experiments are presented and discussed in the ‘Result and Discussion’ section. Results and discussions of the experiments are presented in the respective section. Lastly, the ‘Conclusion’ section summarizes key findings and suggests potential directions for future research.

Related Work

The field of TCR sequencing has gained significant interest in recent years due to its potential applications in immunology, cancer research, and personalized medicine (Pauken et al., 2022). Consequently, there has been a growing body of research focused on developing computational methods for TCR sequence analysis (Finotello et al., 2019). Various algorithms and tools have been proposed for TCR sequence alignment, clustering, and classification. For instance, the MiTCR algorithm was developed to identify TCR alpha and beta chains and determine their variable region sequences (Bolotin et al., 2013). However, accurately identifying and classifying TCR sequences poses challenges due to their high diversity and variable region lengths (Chen et al., 2017; Tillinghast, Behlke & Loh, 1986).

Machine learning approaches have been employed to generate embeddings from TCR sequences and utilize them for classification tasks. For example, the TCRdist algorithm utilizes a k-nearest neighbors approach to cluster TCR sequences based on similarity (Dash et al., 2017). Similarly, the TCRex model employs random forests to predict TCR-major histocompatibility complex (MHC) binding affinity (Gielis et al., 2018). Spike2Vec and PWM2Vec are two additional methods for generating embeddings from protein sequences. Spike2Vec counts the occurrences of consecutive substrings of length K to convert protein sequences into numerical vectors (Ali & Patterson, 2021), while PWM2Vec generates numerical vectors while preserving the position-wise relative importance and ordering information of amino acids (Ali et al., 2022). However, these methods may face challenges in accurately and efficiently performing classification tasks given the complexity and diversity of the dataset.

Deep learning approaches have also been applied to TCR sequencing data for cancer classification. For instance, the DeepTCR model utilizes a deep convolutional neural network (CNN) to classify TCR sequences as cancer-specific or non-cancer-specific (Sidhom et al., 2021). The model is trained on a dataset of TCR sequences from patients to classify different types of cancer (Sidhom & Baras, 2021). Deep learning methods have been used to generate embeddings from protein sequences as well, particularly for predicting protein functions and properties. ProtTrans utilizes a transformer-based neural network to generate embeddings from protein sequences (Elnaggar et al., 2021). Another approach, UniRep, generates fixed-length vector representations of protein sequences for tasks such as protein family classification (Alley et al., 2019) (Bao et al., 2022).

While deep learning methods have shown promise in generating embeddings and performing classification tasks, they can be computationally expensive, especially for large TCR datasets. This computational complexity poses challenges when applying deep learning-based embedding generation methods for protein sequence analysis to large-scale TCR datasets.

Proposed Approach

This section begins with an explanation of the features of TCR protein sequences and how we incorporate them into our embeddings. We then provide a comprehensive overview of the entire pipeline using the flowchart-based approach for each embedding combination method.

Features of TCR protein sequences

The amino acid sequence of a TCR is highly variable, reflecting the diversity of the TCR repertoire that is necessary for effective immune function. Given a TCR (protein) sequence as input, various features can provide insights into its structure, function, and diversity. These features include (i) amino acid sequence composition, (ii) the length of the CDR3 region, (iii) hydrophobicity, (iv) charge distribution, (v) average motif similarity score, (vi) Shannon entropy, and (vii) Simpson index. We will now discuss those features one by one.

Amino acid sequence composition

Amino acid sequence composition in the shape of a numerical vector refers to the frequencies or proportions of the different amino acids in a protein sequence. The numerical vector has a length of 20 because 20 common amino acids can be found in protein sequences. In the context of TCR sequencing data, amino acid sequence composition can be used to characterize the diversity and functional properties of the TCR repertoire (Izraelson et al., 2018). Moreover, the amino acid sequence composition can provide information about the functional properties of the TCR repertoire. For example, certain amino acids may be enriched in TCR sequences that are specific for tumor-associated antigens or other disease-associated antigens (Pan & Li, 2022).

Length of the CDR3 region

The complementarity-determining region 3 (CDR3) is highly variable in the antigen receptor genes of B-cells and T-cells, which is critical for the specificity of antigen recognition. The length of the CDR3 region is typically measured in terms of the number of amino acids, forming a scalar integer.

The length of the CDR3 region can be associated with the affinity and specificity of TCR recognition (Glanville et al., 2017).

Hydrophobicity

In TCR biology, hydrophobicity refers to the tendency of certain amino acid residues in the TCR to interact with hydrophobic residues in the major histocompatibility complex (MHC) molecule and/or the peptide antigen that is presented by the MHC(Chowell et al., 2015). It forms a scalar value represented by a float. Hydrophobic interactions play an essential role in the binding of the TCR to the MHC-peptide complex, and the strength of these interactions can affect the affinity and specificity of the TCR for the antigen (Maruyama et al., 1993).

Charge distribution

Charge distribution in TCR refers to the distribution of charged amino acid residues(i.e., positively charged lysine and arginine and negatively charged aspartic acid and glutamic acid) in the complementarity-determining regions (CDRs) of the TCR (Robbins et al., 2008). The charge distribution in the TCR can affect the binding affinity and specificity of the receptor for the antigen (Davis et al., 1998). Therefore, the charge distribution in the TCR CDRs can provide information about the antigen-binding properties of the TCR repertoire.

Average motif similarity score

The average motif similarity score in TCR shows as a scalar float and refers to the average similarity score between the TCR CDR amino acid sequences and a set of predefined motifs or patterns that are known to be associated with certain antigens or disease (Gupta et al., 2007). The average motif similarity score in TCR can provide information about the antigen specificity of the TCR repertoire. TCRs that have high similarity scores with known antigen motifs may be more likely to recognize and respond to the corresponding antigen (Wadie et al., 2022).

Shannon entropy

Shannon entropy is a measure of diversity or uncertainty in a set of data (Shannon, 1948). In the context of TCR sequencing data, Shannon entropy can be used to determine the frequency of each amino acid at each position in the sequence and then use that information to compute the entropy. A high Shannon entropy implies that multiple different amino acids are observed at a particular position in the TCR protein sequence. This can be an indication of functional flexibility or adaptability, as different amino acids at that position may confer different properties or functions to the TCR protein (Krishna et al., 2020).

Simpson index

The Simpson index is a scalar float value between 0 and 1 and is a measure of diversity commonly used in ecology to quantify the evenness of species abundance in a community (Simpson, 1949). In the context of a single TCR (T-cell receptor) protein sequence, the Simpson index can be used to assess the clonality or dominance of specific amino acids in the sequence. A Simpson index closer to 0 indicates higher diversity or richness (Leinster & Cobbold, 2012), meaning that the amino acids in the sequence are evenly distributed or relatively equal in abundance. Conversely, a Simpson index closer to 1 suggests lower diversity or richness, indicating that a few amino acids are dominant or highly abundant in the sequence (Choudhury et al., 2016).

Shannon entropy and Simpson index are measures of diversity or uncertainty within a sequence or a set of sequences. While they are commonly used to assess diversity across a repertoire or a population of sequences, they can also be computed for individual sequences. It’s important to note that when considering a repertoire or a set of sequences, Shannon entropy and Simpson index can provide a broader picture of diversity within the entire population. However, computing these measures for individual sequences allows for a more detailed analysis at the sequence level.

TCellR2Vec generation

The seven features (discussed above) are used to generate our TCellR2Vec embedding. The dimensionality of all features in our approach is equal to one, except for amino acid sequence compositions, which is 20. Therefore, for each TCR sequence, we generate a numerical feature vector with a length of 26. It’s important to note that each of these features is calculated individually for each TCR sequence, rather than being derived from the entire TCR protein sequence. Algorithm 1 presents the pseudocode for computing TCellR2Vec features and its final embedding. The input for this algorithm is a set of sequences S = {s1, s2, …, sn} where S is the set of sequences, n is the number of sequences, and s1 to sn represent the first sequence to the nth sequence in the sequence set. One sequence is shown as an example in Fig. 2A. The algorithm operates by iterating over each sequence in S and calculating features such as CDR3, aa_comp, hydrophobicity, charge, similarity, Shanon, and Simpson as shown in lines 4 to 10, also shown in Fig. 2B. The first feature, CDR3, is computed by counting the number of unique amino acids in each sequence. The second feature, aa_comp, is generated by analyzing the sequence of amino acids present in each protein sequence and counting the frequency of occurrence of each amino acid, line 5 of Algorithm 1 . For example, if a protein sequence contains 15 amino acids and the amino acid alanine appears four times in the sequence, then the frequency of alanine in that sequence would be 0.26. Similarly, the frequency of occurrence of all the other amino acids is also calculated to generate the aa_comp feature vector. We followed the same procedure to calculate the frequency distribution of each amino acid in the sequence “CASSATGNEQFF”, as depicted in the Fig. 2 and the result is the following: [0, 0.166, 0, 0, 0.083, 0, 0.166, 0, 0.083, 0.083, 0, 0, 0.083, 0.166, 0, 0.083, 0.083, 0, 0, 0].

Figure 2 Overall process for generating a feature embedding of a TCR protein sequence associated with lung cancer using TCellR2Vec.

The similarity score feature is calculated by using the BLOcks SUbstitution Matrix 62 (Blosum62) which is a substitution matrix used in bioinformatics for sequence alignment of protein (Henikoff & Henikoff, 1992). It assigns a score to each possible substitution of one amino acid with another, based on the observed frequency of that substitution in the set of aligned sequences. The value calculated for the specific sequence in Fig. 2B is 1. Finally, to calculate hydrophobicity, charge, Shanon, and Simpson features we used specific libraries. For example, to calculate the hydrophobicity, we used the Kyte-Doolittle hydrophobicity scale, which assigns a score to each amino acid based on its hydrophobicity (Kyte & Doolittle, 1982). The hydrophobicity value calculated for the sequence “CASSATGNEQFF” in Fig. 2 is −0.125. The charge distribution was calculated by counting the number of positively and negatively charged amino acids in each sequence. The charge distribution value computed for the sequence “CASSATGNEQFF” in Fig. 2 is −1.465.

For the Shannon feature, we used the Shannon entropy algorithm, which measures the diversity of amino acids in each sequence. This algorithm takes into account both the frequency of each amino acid and the number of different amino acids present in each sequence. The Simpson diversity feature was calculated using the Simpson index, which measures the probability that two amino acids selected at random from a sequence will be different. This index ranges from 0 to 1, with 1 indicating the highest diversity and 0 indicating no diversity. Figure 2 displays the calculated values for Shannon entropy (3.084) and Simpson diversity (0.875). After calculating these features for each sequence in the input set, the algorithm concatenates all the calculated features to obtain the final embedding vector which is shown in Algorithm 1 , line 11 and Fig. 2C. The resulting embeddings can be merged with baseline embedding generator methods and utilized as an input for classification methods in supervised analysis.

In Algorithm 1 , the loop runs for each sequence in |S|, which has n sequences. Thus, the complexity is O(n). Within each sequence of average length m, finding the CDR3 length and calculating amino acid composition, hydrophobicity/charge scores, charge distribution, and the CDR3 motif all take O(m) time. Since these operations are performed for all n sequences, their overall complexity becomes O(m * n). Loading the Blosum62 matrix has a constant time complexity of O(1) as it’s independent of sequence size. Calculating the similarity score for the motif involves iterating over its length (k), which is much smaller than m, resulting in O(k) complexity. Finally, calculating Shanon Entropy and Simpson Index based on sequence length have a complexity of O(m). Overall, the code’s complexity is dominated by O(m * n) operations due to its dependence on both sequence length and the number of sequences.

_______________________ Algorithm 1 TCellR2Vec____________________________________________________________________      Input:  TCR Sequences S     Output:  TCellR2Vec Embedding ϕ  1:  ϕ ← []                                                ⊳ Initialize Embedding 2D array   2:  for i ← 0 To |S| do  3:       seq ← S[i]   4:       CDR3 ← CDR3LENGHT(seq)   5:       aa_comp ← AMINOACIDCOMPOSITIONS(seq)   6:       hydrophobicity ← HYDROPHOBICITY(seq)   7:       charge ← CHARGEDISTRIBUTION(seq)   8:       similarity ← MOTIFSIMILARITY(seq)   9:       Shannon ← SHANNONINDEX(seq) 10:       Simpson ← SIMPSONINDEX(seq) 11:       Vec ← CONCATENATE(CDR3, aa_comp, hydrophobicity, charge, similar-      ity, Shannon, Simpson) 12:       ϕ.append(Vec) 13:  end for 14:  return ϕ             ⊳ Returning list of embeddings for all TCR sequences

Concatenating TCellR2Vec with baseline methods

After generating TCellR2Vec embedding using the features extracted from the protein sequences, we aggregate (concat) those embeddings with recent embedding methods from the literature to enhance their performance. To this end, the embedding methods used from the literature are (i) one-hot encoding, (ii) Spike2Vec, (iii) PWM2Vec, (iv) spaced K-mers, (v) autoencoder, and (vi) WDGRL. We will now discuss these methods in more detail and the process of their concatenation with TCellR2Vec using a TCR protein sequence associated with lung cancer as an example.

One-hot encoding (OHE) and TCellR2Vec

OHE is a method for generating numerical embeddings of sequences by creating binary vectors for each character. Each binary vector has a size equal to the number of possible characters and assigns a value of 1 to the location corresponding to the character and 0 to all others. The resulting binary vectors are then concatenated to form the final numerical embedding of the sequence (Kuzmin et al., 2020). Figure 3 illustrates a flowchart outlining the process of generating an OHE feature embedding merged with embeddings generated through our feature selection method, TCellR2Vec. First, we generate OHE for each amino acid in the TCR sequence, Figs. 3A–3B, and then combine these vectors together to get the final numerical representation of the TCR sequence, in Fig. 3C. Eventually, we merged them with the numerical vectors of TCellR2Vec, Fig. 3D, and used them as our input of classifiers to classify cancer types.

Figure 3 (A–D) Overall process for generating a feature embedding of a TCR protein sequence associated with lung cancer using OHE and TCellR2Vec.

Spike2Vec and TCellR2Vec

Spike2Vec is a method for converting bio-sequences, such as DNA or protein sequences, into numerical embeddings that can be used in machine learning-based classification tasks. It does so by counting the occurrences of K-mers, which are consecutive substrings of length K that retain the ordering information of the sequence (Ali & Patterson, 2021). We set k equal to 3 in our experiments. The flowchart in Fig. 4 provides an overview of the process of a combination of embeddings generated by Spike2Vec and TCellR2Vec methods. First, in the Spike2Vec part of the approach, the TCR protein sequence is obtained and a set of K-mer, representing subsequences of length k, is generated from the TCR sequence while preserving the order of the sequence, in Figs. 4A to 4B. Next, each K-mer is transformed into a low-dimensional vector using the Spike2Vec embedding method, which learns representations of biological sequences based on cooccurrence patterns in large-scale sequence data, Fig. 4C. Finally, we merged the TCellR2Vec features with the Spike2Vec embeddings to get a single feature vector representation of the TCR sequence, which is used as an input for machine learning models to classify the TCR as associated with cancer target label, Fig. 4D.

Figure 4 (A–D) Overall process for generating a feature embedding of a TCR protein sequence associated with lung cancer using Spike2Vec and TCellR2Vec.

PWM2Vec and TCellR2Vec

PWM2Vec is a technique for obtaining numerical embeddings of biological sequences using the K-mers concept. However, unlike other methods that use K-mer frequencies, PWM2Vec assigns weights to each amino acid in a K-mer and uses these weights to generate the embeddings (Ali et al., 2022). Figure 5 presents a flowchart outlining the steps involved in generating a feature embedding using PWM2Vec combined with embeddings generated through our feature selection method, TCellR2Vec. Initially, the TCR protein sequence is used and a set of K-mers are generated and the k is 3 in our approach, Figs. 5A to 5B. Next, a position frequency matrix (PFM) which shows occurrences of amino acids in each K-mer and subsequently a position probability matrix (PPM) is calculated by dividing the frequency of a character in the column by the sum of that column, Figs. 5C to 5D. Then, to avoid 0 values 0.1 is added to all values of PPM, and by using amino acids frequency tables a position weight matrix (PWM) is deliberated by taking the log-likelihood of each character in each cell of the matrix, Fig. 5E divided by its value in amino acids frequency tables, in Figs. 5E to 5G. Lastly from the PWM2Vec part, we generated the numerical representation of the TCR sequence which shows in Fig. 5H. Finally, we merged both, PWM2Vec and TCellR2Vec embeddings and used them as an input for our classifiers, in Fig. 5I.

Figure 5 (A–G) Overall process for generating a feature embedding of a TCR protein sequence associated with lung cancer using PMW2Vev and TCellR2Vec.

Spaced K-mers and TCellR2Vec

In bioinformatics, generating embeddings using K-mers can be challenging due to the sparsity and high dimensionality of the resulting feature space. To address these issues, spaced K-mers have been introduced, which are non-contiguous substrings of length K called g-mers (Singh et al., 2017). The presented flowchart in Fig. 6 outlines the different stages involved in creating a feature embedding for a TCR protein sequence while using a combination of spaced K-mers and TCellR2Vec embeddings. As a first step, the TCR protein sequence is taken as input for the feature embedding generation process, Fig. 6A. the second step is generating g-mers (we considered g = 9 here) for the TCR sequence due to generating compact feature vectors with reduced sparsity and size, Fig. 6B. From those generated g-mers then we computed K-mers (K = 3) and produced the frequency of those K-mers which represent the numerical vector of spaced K-mers method, in Figs. 6C to 6D. Ultimately, we merged spaced k-mers and TCellR2Vec embeddings and used them as our classification models input, Fig. 6E.

Figure 6 (A–E) Overall process for generating a feature embedding of a TCR protein sequence associated with lung cancer using spaced K-mers and TCellR2Vec.

Autoencoder and TCellR2Vec

The autoencoder approach uses a neural network to obtain numerical features from bio-sequence data. It follows an autoencoder architecture in which the encoder module is optimized to generate the embeddings. The encoder performs a non-linear transformation of the data from space X to a low dimensional numerical feature space Z (Xie, Girshick & Farhadi, 2016). In our experiments, we use a two-layered autoencoder network with an ADAM optimizer and MSE loss function to generate the embeddings. Figure 7 illustrates the process of generating embeddings of a TCR sequence, where we combine the outputs of the autoencoder method and TCellR2Vec. Initially, the TCR sequence is taken as input, and for each amino acid in the sequence, the OHE vector is computed, as shown in Figs. 7A and 7B. Subsequently, the OHE vectors are combined to generate the input for a two-layer autoencoder network that includes an encoder and a decoder to generate a low-dimensional numerical representation of the TCR sequence, as depicted in Fig. 7D. Finally, we combine the autoencoder output with TCellR2Vec embeddings to create the final embedding, which serves as input for the classification models, as shown in Fig. 7E.

Figure 7 (A–E) Overall process for generating a feature embedding of a TCR protein sequence associated with lung cancer using autoencoder and TCellR2Vec.

Wasserstein distance guided representation learning (WDGRL) and TCellR2Vec

The WDGRL approach utilizes a neural network to extract numerical features by optimizing the Wasserstein distance (WD) between source and target distributions, making it an unsupervised domain adaptation technique (Shen et al., 2018). Figure 8 illustrates the process of embedding generation for a TCR protein sequence using WDGRL and the TCellR2Vec approach. The first step involved converting each amino acid in the sequence into separate one-hot encoded vectors, as shown in Figs. 8A and 8B. In the next step, we combined all the numerical vectors to form a single vector, which was then used as an input for the WDGRL network. This network was used to extract embeddings by optimizing the WD between the source and target features, as depicted in Fig. 8D. Finally, we combined the results of the WDGRL and TCellR2Vec embeddings to produce a final embedding, which served as the input for the classifier models, as shown in Fig. 8E.

Figure 8 (A–E) Overall process for generating a feature embedding of a TCR protein sequence associated with lung cancer using WDGRL and TCellR2Vec.

Experimental setup

In this section, we provide information about the dataset and data visualization techniques. For the classification task, we employed various ML classifiers, including support vector machine (SVM), naive Bayes (NB), multi-layer perceptron (MLP), K-nearest neighbors (KNN) with K = 3, random forest (RF), logistic regression (LR), and decision tree (DT). A stratified sampling-based 70–30% train-test split was applied, with 10% of the training set reserved for hyperparameter tuning. We repeat experiments with 5 random splits and report average results. Evaluation metrics such as accuracy, precision, recall, weighted F1 score, macro F1 score, ROC-AUC, and classifiers training runtime were used to assess the performance of the baseline models and their combination with our proposed method. For the multi-class classification task, the one-vs-rest approach was utilized to convert binary metrics to multi-class metrics. The choice of evaluation metrics was driven by the need to comprehensively assess the performance of TCellR2Vec and the baseline methods across multiple dimensions, including predictive accuracy, precision, recall, and computational efficiency. Additionally, we aimed to evaluate the methods’ ability to handle the multi-class nature of the cancer classification task. Accuracy is a fundamental metric that provides an overall measure of the correctness of predictions. However, in multi-class classification problems, accuracy alone may not provide a complete picture of performance, as it does not account for class imbalances or the ability to distinguish between different classes. Therefore, we included precision, recall, and F1 scores (both weighted and macro-averaged) to assess the methods’ performance in identifying each cancer type correctly.

Precision measures the proportion of true positives among the positive predictions, indicating the method’s ability to avoid false positives. Recall quantifies the proportion of actual positives that are correctly identified, reflecting the method’s ability to detect all instances of a particular class. The F1 score combines precision and recall into a single metric, providing a balanced measure of performance.

Furthermore, we employed the weighted F1 score to account for class imbalances, ensuring that the performance of minority classes is appropriately weighted. Conversely, the macro-averaged F1 score treats all classes equally, providing an unbiased assessment of performance across all classes, irrespective of their sample sizes. The ROC-AUC (area under the receiver operating characteristic curve) metric was included to evaluate the methods’ ability to discriminate between different classes. ROC-AUC is particularly useful in multi-class classification tasks, as it provides a comprehensive measure of the trade-off between true positive rate and false positive rate across all classes. Finally, training runtime was considered an essential metric to assess the computational efficiency of the methods. In real-world applications, where large-scale TCR sequence analysis is required, computational efficiency can be a critical factor in determining the feasibility and scalability of the approach.

By considering this diverse set of evaluation metrics, we aimed to provide a comprehensive and multi-faceted assessment of TCellR2Vec and the baseline methods. This approach allowed us to evaluate not only the predictive performance but also the ability to handle class imbalances, discriminative power, and computational efficiency, which are all crucial factors in the context of TCR sequence analysis and cancer classification.

The experiments were conducted on a computer system with an Intel(R) Core i5 processor, 32 GB of memory, and a 64-bit Windows 10 operating system. The models were implemented in Python. Our preprocessed data and code is available online for reproducebility (http://www.github.com/zara77/TcellR2Vec.git).

Dataset statistics

We obtained our TCR beta chain sequence data from TCRdb, a comprehensive database for T-cell receptor sequences that offers a powerful search function (Chen et al., 2021) with more than 277 million sequences collected from over 8265 TCR-Seq samples derived from hundreds of tissues, clinical conditions, and cell types. This study focused on identifying and extracting data on the five most prevalent types of cancer, as determined by their incidence rates. To extract a subset from the original data while preserving the distribution of target labels(cancer types), we used the Stratified ShuffleSplit method and randomly extracted 50,731 TCR sequences for five different types of cancer (see Table 1). The Sequence Length Statistics in Table 1 indicate that the average sequence length is 15 for most cancer types, highlighting the challenge of working with sequences of very short lengths. Total unique TCR sequences compared with the total number of sequences for each cancer type shows the diversity of this dataset which is due to the importance of the uniqueness of the TCR sequences for the immune system’s ability to recognize and respond to a wide variety of pathogens.

Table 1 Dataset statistics of TCR sequences.

The table shows the total number and the unique number of sequences for each cancer type, the minimum, the average, and the maximum length of TCR sequences in the dataset used for the experiments in this study.

			Sequence length statistics	
Cancer type	Total sequences	Unique sequences	Min.	Max.	Average	
Glioblastoma	13,970	13,543	6	28	14	
Lung	12,616	12,065	7	30	15	
Melanoma	17,063	11,303	8	25	15	
Osteosarcoma	1,453	1,453	7	24	15	
Pancreatic	5,629	5,617	11	25	15	
Total	50,731	43,981	–	–	–	

Table 2 provides examples of TCR sequences, cancer names, and gene mutations for Five different cancer types. The “Gene Mutation” column presents the tumor suppressor gene mutations that are linked to an elevated risk of developing these cancers. For instance, BRAF and EGFR mutations are linked to a higher likelihood of developing lung cancer.

Table 2 An example of sequences for five different cancer types, glioblastoma, lung, melanoma, pancreatic, and osteosarcoma along with their respective gene mutations.

Sequence	Cancer name	Gene mutation	Reference	
CSATGSSYNEQFF	Glioblastoma	EGFR, TP53, PIK3CA, NF1	Aldape et al. (2015)	
CSAPGTNYNEQFF	Lung	BRAF, KRAS, ALK, EGFR, ROS1	Li, Qu & Xu (2015)	
CATSSGNTIYF	Melanoma	TP53, CDKN2A, NRAS, BRAF	Daniotti et al. (2004)	
CASRRTGRNQPQHF	Pancreatic	KRAS,TP53, CDKN2A, BRCA1/BRCA2	Waddell et al. (2015)	
CSVKKGAGNTIYF	Osteosarcoma	MYC,TP53, RB1, PIK3CA, CDKN2A	Perry et al. (2014)	

Data visualization

To explore the data visually, we employed t-distributed stochastic neighbor embedding (t-SNE) to map the input sequences into a 2D representation (Van der Maaten & Hinton, 2008). Figures 9A to 9F shows the t-SNE results for different embedding methods, including one-hot-encoding (OHE), Spike2Vec, PWM2Vec, spaced K-mer, autoencoder, and TCellR2Vec (our embeddings). Our observations indicate that OHE, Spike2Vec, PWM2Vec, and autoencoder exhibit a smaller number of groups for different cancer types, whereas spaced K-mer shows a more scattered pattern. Our method in Fig. 9F provides a cohesive representation, resulting in a cleaner overall structure. Additionally, we provided t-SNE plots for merging embeddings of different baseline methods with TCellR2Vec embeddings, as shown in Fig. 9G to 9K. The same pattern is repeated here, except for autoencoder, which shows a more scattered pattern. Overall, this improved representation enhances the interpretability and effectiveness of the results, facilitating a better understanding of the underlying patterns and relationships in the data.

Figure 9 (A–K) tSNE plots for different types of baseline embedding generation methods, TCellR2Vec embeddings, and combination of baselines with TCellR2Vec.

Results And Discussion

In this section, we present the classification results for both baseline methods and their aggregation with our proposed method using various evaluation metrics. The findings of this study unequivocally demonstrate the remarkable efficacy of the proposed TCellR2Vec method in significantly enhancing the classification performance of TCR sequences for accurately identifying different cancer types. By ingeniously combining meticulously selected informative features extracted from TCR sequences with state-of-the-art baseline embedding methods, TCellR2Vec introduces a powerful and comprehensive representation that adeptly captures the most pertinent characteristics crucial for achieving superior cancer classification accuracy. Table 3 presents a comparison of the classification performance of different baseline methods. We can observe that OHE and Spike2Vec utilizing RF classifier, outperform other methods including our proposed method (TCellR2Vec) in terms of evaluation metrics except for the training runtime metric. Considering the training runtime metric, our method outperforms all baseline methods. Although there is only a small difference (around 2%) in accuracy between TCellR2Vec and OHE, our proposed approach exhibited a distinct competitive edge by demonstrating remarkable efficiency in training runtime. This computational efficiency is an invaluable asset for real-world applications where time and resource constraints are critical factors. Additionally, it makes our method suitable for use with large datasets. Comparing TCellR2Vec with PWM2Vec, autoencoder, and WDGRL shows our method outperformed these baseline methods in terms of predictive performance.

Table 3 Classification results (averaged over five runs) for different evaluation metrics.

The best values are shown in bold.

Embeddings	Algo.	Acc. ↑	Prec. ↑	Recall ↑	F1 (Weig.) ↑	F1 (Macro) ↑	ROC AUC ↑	Train Time (sec.) ↓	
OHE	SVM	0.4289	0.4049	0.4289	0.3291	0.2246	0.5549	159188.43	
NB	0.0496	0.3978	0.0496	0.2612	0.0430	0.5059	5.0369	
MLP	0.4152	0.3783	0.4152	0.3657	0.2697	0.5593	722.7711	
KNN	0.3934	0.3650	0.3934	0.3613	0.2682	0.5544	48.9053	
RF	0.4779	0.4343	0.4779	0.4230	0.3080	0.5888	73.2682	
LR	0.4304	0.3885	0.4304	0.3525	0.2477	0.5597	518.7145	
DT	0.4465	0.4286	0.4465	0.4089	0.3234	0.5874	9.4110	
Spike2Vec	SVM	0.4054	0.3571	0.4054	0.3073	0.2000	0.5388	746.112	
NB	0.3654	0.3502	0.3654	0.3347	0.2514	0.5472	0.0790	
MLP	0.4166	0.3763	0.4166	0.3587	0.2430	0.5523	24.0041	
KNN	0.3921	0.3661	0.3921	0.3722	0.2659	0.5536	14.0131	
RF	0.4710	0.4297	0.4710	0.4404	0.3118	0.5874	36.5958	
LR	0.4202	0.3748	0.4202	0.3516	0.2345	0.5511	1.7132	
DT	0.4303	0.4226	0.4303	0.4256	0.3143	0.5811	4.3699	
PWM2Vec	SVM	0.4015	0.4079	0.4015	0.2936	0.1898	0.5350	899.8259	
NB	0.3375	0.3378	0.3375	0.2989	0.2303	0.5368	0.0823	
MLP	0.4031	0.3598	0.4031	0.3216	0.2114	0.5412	31.6471	
KNN	0.3791	0.3510	0.3791	0.3605	0.2570	0.5463	23.6907	
RF	0.4545	0.4139	0.4545	0.4241	0.3002	0.5792	90.4235	
LR	0.4018	0.3958	0.4018	0.3158	0.2072	0.5386	4.3419	
DT	0.4152	0.4065	0.4152	0.4099	0.3053	0.5743	8.4342	
Spaced K-mers	SVM	0.3857	0.2716	0.3857	0.2878	0.1860	0.5316	645.3608	
NB	0.2843	0.3340	0.2843	0.2583	0.2153	0.5366	0.0471	
MLP	0.3980	0.3411	0.3980	0.3234	0.2137	0.5397	23.5225	
KNN	0.3908	0.3624	0.3908	0.3687	0.2620	0.5519	11.0880	
RF	0.4610	0.4204	0.4610	0.4325	0.3068	0.5833	35.0048	
LR	0.3905	0.3352	0.3905	0.3036	0.1984	0.5343	1.6265	
DT	0.4189	0.4091	0.4189	0.4136	0.3030	0.5734	3.8643	
Autoencoder	SVM	0.4009	0.3270	0.4009	0.3083	0.2037	0.5417	942.0830	
NB	0.3878	0.3475	0.3878	0.3299	0.2319	0.5423	0.0581	
MLP	0.4074	0.3465	0.4074	0.3472	0.2347	0.5485	29.9132	
KNN	0.3890	0.3561	0.3890	0.3661	0.2612	0.5517	26.3019	
RF	0.4667	0.4175	0.4667	0.4126	0.2832	0.5795	82.0532	
LR	0.4081	0.3385	0.4081	0.3211	0.2135	0.5461	2.2471	
DT	0.4252	0.4134	0.4252	0.4181	0.3117	0.5792	6.8419	
WDGRL	SVM	0.3350	0.1122	0.3350	0.1681	0.1004	0.5000	235.7632	
NB	0.3382	0.4161	0.3382	0.2481	0.1695	0.5128	0.0609	
MLP	0.3690	0.3493	0.3690	0.3407	0.2411	0.5363	74.9861	
KNN	0.3427	0.3235	0.3427	0.3278	0.2359	0.5291	2.5699	
RF	0.4255	0.3871	0.4255	0.3923	0.2746	0.5632	27.8623	
LR	0.3447	0.2132	0.3447	0.2313	0.1462	0.5085	0.9741	
DT	0.3956	0.3854	0.3956	0.3892	0.2917	0.5650	1.0738	
TCellR2Vec (Ours)	SVM	0.3892	0.2428	0.3892	0.2935	0.1888	0.5331	139.6315	
NB	0.3766	0.3258	0.3766	0.3185	0.2160	0.5335	0.0234	
MLP	0.3909	0.3178	0.3909	0.3208	0.2110	0.5360	18.6838	
KNN	0.3595	0.3333	0.3595	0.3401	0.2410	0.5364	10.7244	
RF	0.4574	0.3999	0.4574	0.4168	0.2835	0.5758	15.9207	
LR	0.3877	0.3037	0.3877	0.3015	0.1956	0.5338	0.2061	
DT	0.4057	0.3968	0.4057	0.4003	0.2921	0.5665	1.1230	

Furthermore, we compared baseline methods while combining them with our proposed method, Table 4, and the results demonstrate that resoundingly integrating TCellR2Vec with baseline methods such as Spike2Vec, PWM2Vec, spaced K-mers and Autoencoder consistently yielded substantial performance improvements across a comprehensive array of evaluation metrics. The observed improvements, ranging from approximately 1–2% in accuracy, precision, recall, F1 score, and ROC AUC, unequivocally underscore the synergistic potential of TCellR2Vec and its unparalleled ability to augment and fortify existing embedding techniques.

Table 4 Classification results for merging baseline embeddings and TCellR2Vec embeddings (averaged over five runs) for different evaluation metrics.

The best values are shown in bold.

Embeddings	Algo.	Acc. ↑	Prec. ↑	Recall ↑	F1 (Weig.) ↑	F1 (Macro) ↑	ROC AUC ↑	Train time (sec.) ↓	
TCellR2Vec + OHE	SVM	0.4082	0.3499	0.4082	0.3108	0.2031	0.5411	1527.89	
NB	0.4008	0.3439	0.4008	0.3322	0.2223	0.5435	0.0596	
MLP	0.4154	0.3746	0.4154	0.3518	0.2360	0.5502	17.5448	
KNN	0.3901	0.3579	0.3901	0.3677	0.2603	0.5510	11.4929	
RF	0.4740	0.4231	0.4740	0.4231	0.2897	0.5825	33.3112	
LR	0.4119	0.3308	0.4119	0.3244	0.2138	0.5463	1.2806	
DT	0.4280	0.4188	0.4280	0.4224	0.3129	0.5798	3.4360	
TCellR2Vec + Spike2Vec	SVM	0.4358	0.4028	0.4358	0.3397	0.2229	0.5563	1155.2585	
NB	0.3823	0.3583	0.3823	0.3541	0.2613	0.5524	0.0556	
MLP	0.4363	0.3950	0.4363	0.3826	0.2598	0.5627	39.1359	
KNN	0.3998	0.3705	0.3998	0.3785	0.2704	0.5565	21.9391	
RF	0.4901	0.4416	0.4901	0.4472	0.3095	0.5923	56.4536	
LR	0.4380	0.3825	0.4380	0.3640	0.2424	0.5604	2.7763	
DT	0.4396	0.4326	0.4396	0.4354	0.3212	0.5857	7.4239	
TCellR2Vec + PWM2Vec	SVM	0.4248	0.3979	0.4248	0.3293	0.2171	0.5519	556.2903	
NB	0.3453	0.3375	0.3453	0.3153	0.2384	0.5395	0.0624	
MLP	0.4254	0.3740	0.4254	0.3452	0.2306	0.5554	15.7060	
KNN	0.3939	0.3620	0.3939	0.3717	0.2668	0.5538	12.2160	
RF	0.4758	0.4256	0.4758	0.4333	0.3012	0.5866	31.2911	
LR	0.4279	0.4045	0.4279	0.3451	0.2304	0.5556	1.3041	
DT	0.4248	0.4151	0.4248	0.4191	0.3090	0.5779	3.2203	
TCellR2Vec + Spaced K-mers	SVM	0.4065	0.3656	0.4065	0.3093	0.2012	0.5424	342.856	
NB	0.2945	0.3310	0.2945	0.2678	0.2215	0.5378	0.1191	
MLP	0.4274	0.3984	0.4274	0.3437	0.2280	0.5543	26.7004	
KNN	0.3770	0.3488	0.3770	0.3570	0.2533	0.5450	12.2576	
RF	0.4696	0.4233	0.4696	0.4304	0.3004	0.5837	31.5703	
LR	0.4163	0.3290	0.4163	0.3287	0.2158	0.5485	1.4139	
DT	0.4267	0.4180	0.4267	0.4216	0.3121	0.5792	3.8184	
TCellR2Vec + Autoencoder	SVM	0.4079	0.3376	0.4079	0.3150	0.2065	0.5433	1467.6195	
NB	0.3901	0.3466	0.3901	0.3307	0.2273	0.5410	0.0531	
MLP	0.4150	0.3931	0.4150	0.3471	0.2335	0.5494	27.9985	
KNN	0.3906	0.3590	0.3906	0.3689	0.2617	0.5515	12.2362	
RF	0.4727	0.4155	0.4727	0.4200	0.2859	0.5811	33.3520	
LR	0.4087	0.3286	0.4087	0.3231	0.2130	0.5449	1.1202	
DT	0.4236	0.4148	0.4236	0.4184	0.3109	0.5783	3.4886	
TCellR2Vec + WDGRL	SVM	0.3488	0.2140	0.3488	0.2177	0.1331	0.5050	71.0521	
NB	0.3514	0.2961	0.3514	0.2572	0.1638	0.5121	0.0359	
MLP	0.3603	0.3363	0.3603	0.3259	0.2290	0.5292	20.6209	
KNN	0.3296	0.3147	0.3296	0.3159	0.2299	0.5236	1.7007	
RF	0.4153	0.3765	0.4153	0.3824	0.2648	0.5570	9.9161	
LR	0.3551	0.2181	0.3551	0.2517	0.1585	0.5120	0.1929	
DT	0.3846	0.3768	0.3846	0.3799	0.2792	0.5572	0.4387	

Upon comparing the evaluation metrics, we found that the combination of TCellR2Vec with OHE and WDGRL did not show any improvement in comparison to using OHE and WDGRL methods alone, except for the training runtime metric which shows an enhancement. Since Table 1 (“Unique Sequences” column) elaborates on the diversity of our dataset, it is evident that the similarity between sequences for different cancer types is very low. As a result, the underlying classifiers face a difficult time in differentiating between them. However, even with such a challenging dataset, merging baselines with TCellR2Vec has led to improvement in the predictive performance of ML classifiers, which is evidence of the effectiveness of our method. The findings of this study highlight the potential of TCellR2Vec as an efficient and effective feature selection method for TCR sequence analysis in cancer classification. However, the applications and utilities of TCellR2Vec extend beyond cancer classification alone. TCellR2Vec can play a pivotal role in the development of personalized cancer immunotherapies by aiding in the identification of tumor-reactive T-cell receptors and neoantigens, paving the way for the development of personalized cancer vaccines and adoptive cell transfer therapies tailored to individual patients’ immune profiles. Additionally, TCellR2Vec can be employed to monitor the patient’s immune response during cancer treatment, providing insights into the effectiveness of immunotherapies and guiding treatment decisions. The ability of TCellR2Vec to capture informative features from TCR sequences could also facilitate the discovery of novel biomarkers associated with cancer prognosis, treatment response, or immune system dysfunction. Furthermore, TCellR2Vec embeddings could be integrated with neoantigen prediction algorithms to improve the identification of tumor-specific neoantigens and enhance the development of personalized cancer vaccines. By analyzing the diversity and characteristics of TCR sequences associated with different cancer types, TCellR2Vec can contribute to a better understanding of tumor-immune interactions and the mechanisms underlying immune evasion by cancer cells.

Statistical significance

To ensure the credibility and reliability of the classification results, we performed p-value calculations for each method. These calculations were based on the average and standard deviation (the std. values are not included in the paper due to page limitation) of all evaluation metrics, obtained from five experiment runs. Notably, the p-values for all comparisons between the proposed model and the baselines were found to be less than the significance level of 0.05. However, it is important to highlight that the training runtime metric exhibited different behavior, revealing higher variability in the training runtime values, which led to certain p-values exceeding the 0.05 threshold. This discrepancy can be attributed to several factors, such as processor performance and the number of concurrent processes at any given time, which can impact the training time.

Statistical analysis

One approach to assess the efficacy of the feature embeddings is through an analysis of their compactness. To accomplish this, we conduct statistical analyses, specifically employing Pearson and Spearman correlation measures. These measures allow us to calculate the correlation values between the various features of the embeddings, which correspond to the target labels (cancer types). We then determine the proportion of attributes within each feature embedding that exhibit a strong correlation with the class labels. The Pearson correlation values for different thresholds are presented in Fig. 10, while Fig. 10 displays the Spearman correlation values for different thresholds spanning from −1 to 1, across multiple embeddings. We can observe that although WDGRL shows a higher correlation within the range < − 0.1 and >0.1, the autoEncoder and TCellR2Vec (only sequence features) show a comparable behavior. This shows that with only features (and not sequences) in the embeddings, TCellR2Vec is able to capture a correlation of features with the cancer-type labels that is comparable to the correlations of WDGRL and autoEncoder from the baseline. Compared to embedding methods like Spike2Vec and PWM2Vec, the TCellR2Vec’s embedding features are highly correlated with the class label, demonstrating that this embedding is more compact, hence preserving more valuable information.

Figure 10 Correlation values for the T-Cell dataset where (A) and (B) show the fraction of features having correlation values greater than or less than the thresholds (on the x-axis).

The fractions are computed by taking the denominator as the size of the embeddings.

Conclusion

In this study, we proposed TCellR2Vec, a novel feature selection method for TCR sequences in cancer classification. Our approach focuses on identifying informative features such as amino acid sequence compositions and CDR3 length, etc., and merging them with baseline methods’ embedding vectors for supervised analysis. We observed improved predictive performance in TCR-based cancer classification models. This demonstrates the effectiveness and suitability of our proposed method for the task at hand. Future work can explore the incorporation of additional features and the utilization of other unsupervised learning methods to further enhance the performance of TCellR2Vec. Furthermore, the applicability of TCellR2Vec to other TCR datasets, such as different cancer types or autoimmune disease datasets, warrants investigation. Integration of TCellR2Vec with other bioinformatics tools, such as those for neoantigen identification or TCR binding affinity prediction, could provide a more comprehensive analysis of TCR sequences and their implications.

Supplemental Information

Supplemental Information 1 Labels.

Supplemental Information 2 Dataset.

Additional Information and Declarations

Competing Interests

Author Contributions

Data Availability

The authors declare there are no competing interests.

Zahra Tayebi conceived and designed the experiments, performed the experiments, analyzed the data, performed the computation work, prepared figures and/or tables, authored or reviewed drafts of the article, and approved the final draft.

Sarwan Ali conceived and designed the experiments, performed the experiments, analyzed the data, performed the computation work, prepared figures and/or tables, authored or reviewed drafts of the article, and approved the final draft.

Murray Patterson conceived and designed the experiments, authored or reviewed drafts of the article, and approved the final draft.

The following information was supplied regarding data availability:

The code and data are available at GitHub and Zenodo:

- https://github.com/zara77/TcellR2Vec

- Zahra, T. (2024). TCellR2Vec Dataset and Code. In TCellR2Vec: Efficient feature selection for TCR sequences for cancer classification. Zenodo. https://doi.org/10.5281/zenodo.12167095.

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
