# Peer review of "TCellR2Vec: efficient feature selection for TCR sequences for cancer classification"

_PeerJ Computer Science, doi:10.7717/peerj-cs.2239_

## Round 0.1 · original submission · Minor Revisions

The reviewers find merit in your paper and have suggested a few changes. As an Associate Editor, I also recommend the following:

(1) In the proposed methodology section, it would be better if a workflow diagram were introduced to help the reader understand the methodology easily.
(2) The discussion part needs to be strengthened.

Reviewer 2 has suggested that you cite specific references. You are welcome to add it/them if you believe they are relevant. However, you are not required to include these citations, and if you do not include them, this will not influence my decision.

You are required to revise your manuscript and resubmit it.

**Language Note:** The review process has identified that the English language must be improved. PeerJ can provide language editing services - please contact us at [email protected] for pricing (be sure to provide your manuscript number and title). Alternatively, you should make your own arrangements to improve the language quality and provide details in your response letter. – PeerJ Staff

Reviewer 1 ·

Basic reporting

Language improvement needed.
Extend introduction section by adding latest updates about the Cancer and their treatment process.
The definitions of certain terms used in the equations and theorems are missing from the Formal findings,
It is strongly advised that appropriate terminology and meanings be added to these formulas.

Experimental design

A more thorough explanation of the methodology is needed to help the reader in understanding the different machine learning algorithms used in generating models.
Several new citations and references from recent research articles should be included in the manuscript.

Validity of the findings

Highlight the applications and utility of the work.
Overall, this is an interesting study and results obtained are good.

Reviewer 2 ·

Basic reporting

Authors have proposed a TCellR2Vec system which seems to be a promising approach to tackle the challenge of feature selection in TCR sequence analysis for cancer classification

Experimental design

Clarifying the rationale behind the choice of evaluation metrics . This would enhance the understanding of the performance comparison between TCellR2Vec and baseline methods.
Authors need to Explore the impact of dataset characteristics, such as sample size and class distribution, on the performance of TCellR2Vec could provide further insights into its generalizability.
Being an interdisciplinary study may cite the following PMID: 35215090,PMID: 34829338

What were the potential limitations or challenges encountered during the development and evaluation of TCellR2Vec. This would enrich the discussion and provide a balanced perspective.
May provide visualizations or examples of feature embeddings generated by TCellR2Vec could aid in understanding its effectiveness in capturing informative aspects of TCR sequences.

Validity of the findings

Elaborate on the computational complexity of TCellR2Vec.
Discuss its scalability to larger datasets, which would address potential concerns regarding its practical implementation.
Discussing the interpretability of feature embeddings. How they contribute to the overall classification performance would deepen the understanding of functionality of the proposed system.

Additional comments

Note: English and grammar needs improvements

---

## Round 0.2 · accepted · Accept

I am pleased to inform you that your paper has been accepted for publication in PeerJ Computer Science. Your manuscript has undergone rigorous peer review, and I am delighted to say that it has been met with praise from our reviewers and editorial team. Your research makes a significant contribution to the field, and we believe it will be of great interest to our readership. On behalf of the editorial board, I extend our warmest congratulations to you.

Reviewer 1 ·

Basic reporting

I enjoyed reading this manuscript and believe that it is very promising.
Overall, this is an interesting study, and the results obtained are good.

Experimental design

The authors have improved the manuscript as per my previous suggestions. No new suggestions. The manuscript can be considered for acceptance.

Validity of the findings

This method is a reliable, rapid and useful prediction method. No new suggestions. The manuscript can be considered for acceptance.

Additional comments

The manuscript has been revised as per my comments and suggestions.

Reviewer 2 ·

Basic reporting

no comment

Experimental design

no comment

Validity of the findings

no comment